# Estimating PM$_{10}$ Concentration from Drilling Operations in Open-Pit Mines Using an Assembly of SVR and PSO

**Xuan-Nam Bui [1,2,*]**, **Chang Woo Lee [3]**, **Hoang Nguyen [4,*]**, **Hoang-Bac Bui [5,6]**,
**Nguyen Quoc Long [7]**, **Qui-Thao Le [1,2]**, **Van-Duc Nguyen [3]**, **Ngoc-Bich Nguyen [8]**
**and Hossein Moayedi [9,10,*]**

[1] Department of Surface Mining, Mining Faculty, Hanoi University of Mining and Geology, Duc Thang, Bac Tu Liem, Hanoi 100000, Vietnam

[2] Center for Mining, Electro-Mechanical Research, Hanoi University of Mining and Geology, Duc Thang, Bac Tu Liem, Hanoi 100000, Vietnam

[3] Department of Energy and Mineral Resources, College of Engineering, Dong-A University, Busan 49315, Korea

[4] Institute of Research and Development, Duy Tan University, Da Nang 550000, Vietnam

[5] Faculty of Geosciences and Geoengineering, Hanoi University of Mining and Geology, 18 Vien st., Duc Thang Ward, Bac Tu Liem dist., Hanoi 100000, Vietnam

[6] Center for Excellence in Analysis and Experiment, Hanoi University of Mining and Geology, 18 Vien st., Duc Thang Ward, Bac Tu Liem dist., Hanoi 100000, Vietnam

[7] Department of Mine Surveying, Hanoi University of Mining and Geology, Duc Thang, Bac Tu Liem, Hanoi 100000, Vietnam

[8] Faculty of Environmental and Occupational Health, Hanoi University of Public Health, 1A Duc Thang Road, Duc Thang Ward, North Tu Liem District, Hanoi 100000, Vietnam

[9] Department for Management of Science and Technology Development, Ton Duc Thang University, Ho Chi Minh City 700000, Vietnam

[10] Faculty of Civil Engineering, Ton Duc Thang University, Ho Chi Minh City 700000, Vietnam

\* Correspondence: buixuannam@humg.edu.vn (X.-N.B.); nguyenhoang23@duytan.edu.vn (H.N.); hossein.moayedi@tdtu.edu.vn (H.M.)

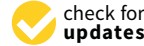

**Featured Application: This study provided a new artificial intelligence system (i.e., PSO-SVR) to predict and control PM$_{10}$ concentration induced by drilling operations in open-pit mines. By the use of this system, the air quality can be managed as a part of the whole air quality in open-pit mines. Also, occupational diseases can be controlled and minimized by this system.**

**Abstract:** Dust is one of the components causing heavy environmental pollution in open-pit mines, especially PM$_{10}$. Some pathologies related to the lung, respiratory system, and occupational diseases have been identified due to the effects of PM$_{10}$ in open-pit mines. Therefore, the prediction and control of PM$_{10}$ concentration in the production process are necessary for environmental and health protection. In this study, PM$_{10}$ concentration from drilling operations in the Coc Sau open-pit coal mine (Vietnam) was investigated and considered through a database including 245 datasets collected. A novel hybrid artificial intelligence model was developed based on support vector regression (SVR) and a swarm optimization algorithm (i.e., particle swarm optimization (PSO)), namely PSO-SVR, for estimating PM$_{10}$ concentration from drilling operations at the mine. Polynomial (P), radial basis function (RBF), and linear (L) kernel functions were considered and applied to the development of the PSO-SVR models in the present study, abbreviated as PSO-SVR-P, PSO-SVR-RBF, and PSO-SVR-L. Also, three benchmark artificial intelligence techniques, such as *k*-nearest neighbors (KNN), random forest (RF), and classification and regression trees (CART), were applied and developed for estimating PM$_{10}$ concentration and then compared with the PSO-SVR models. Root-mean-squared error (RMSE)

and determination coefficient ($R^2$) were used as the statistical criteria for evaluating the performance of the developed models. The results exhibited that the PSO algorithm had an essential role in the optimization of the hyper-parameters of the SVR models. The PSO-SVR models (i.e., PSO-SVR-L, PSO-SVR-P, and PSO-SVR-RBF) had higher performance levels than the other models (i.e., RF, CART, and KNN) with an RMSE of 0.040, 0.042, and 0.043; and $R^2$ of 0.954, 0.948, and 0.946; for the PSO-SVR-L, PSO-SVR-P, and PSO-SVR-RBF models, respectively. Of these PSO-SVR models, the PSO-SVR-L model was the most dominant model with an RMSE of 0.040 and $R^2$ of 0.954. The remaining three benchmark models (i.e., RF, CART, and KNN) yielded a more unsatisfactory performance with an RMSE of 0.060, 0.052, and 0.067; and $R^2$ of 0.894, 0.924, and 0.867, for the RF, CART, and KNN models, respectively. Furthermore, the findings of this study demonstrated that the density of rock mass, moisture content, and the penetration rate of the drill were essential parameters on the $PM_{10}$ concentration caused by drilling operations in open-pit mines.

**Keywords:** meta-heuristic algorithm; $PM_{10}$ concentration; drilling operation; artificial intelligence; open-pit coal mine

## 1. Introduction

Open-pit mining is one of the technologies that aim to pick up natural resources underground. However, the impact from the operations of open-pit mines, such as drilling, blasting, loading, transporting, and dumping are significant [1]. Also, a large area of land is occupied for work and dump sites. Other impacts on the property, water, and atmosphere due to open-pit mining activities are also considered to be significant [2–4]. Of those, dust concentration has a significant impact on public health and environmental pollution. The particles in the dust with an equivalent aerodynamic diameter smaller than 10 μm ($PM_{10}$) are considered as a hazardous factor for atmospheric and public health [5–7]. This dust is cause-related to lung, respiratory, and ocular diseases [3,8]. The fauna and flora of the surrounding area are also affected by the dust caused by open-pit mining operations [9,10]. In open-pit mines, many activities raise $PM_{10}$ dust, such as transporting, drilling-blasting, loading/unloading, and dumping [11–16]. These activities can be divided into point, line, and area sources [17,18]. Of those activities, drilling is, at 25% of the whole operation, one of the point sources that generates dust in the environment and is one of particular concern. Predicting and controlling $PM_{10}$ concentration in each operation is the basis of the development of the total predictive model for primary activities in open-pit mines (i.e., drilling, blasting, loading/unloading, transporting). Accurate prediction and strict control of $PM_{10}$ concentration from drilling operations in open-pit mines are essential to ensuring health and atmospheric safety, as well as providing the significant basis for the total predictive model.

According to the United States Environmental Protection Agency (USEPA), dust concentration and emission can be predicted by various empirical equations. Several scholars have also proposed experimental equations to estimate dust concentration [19–24]. However, those equations were not suitable for mining conditions in Vietnam, especially deep mines [25,26]. In open-pit mines with deep mining, the problem of natural ventilation is often complicated due to the impact of mining depth because dust and toxic gases are often not circulated, making it challenging to calculate dust concentration as well as poisonous gases. Ghose [27] proposed several equations for predicting dust concentration in an open-pit coal mine of India; however, dust concentration from drilling operations was not considered in his study. Meanwhile, drilling operations are one of the main factors causing environmental pollution in open-pit mines, especially $PM_{10}$ emission.

In recent years, quantitative models have been proven as effective methods to predict environmental issues and control atmospheric pollution. Artificial intelligence (AI) and its applications were considered as the robust tools for predicting and controlling environmental issues, especially in mining operations [28–44]. For estimating $PM_{10}$ concentration, Chelani and Gajghate [45] used an artificial

neural network (ANN) based on the back-propagation (BP) algorithm to predict $PM_{10}$ concentration. The feasibility of ANN was interpreted in their study with promising results. By a similar approach, McKendry [46] also developed an ANN model to predict $PM_{10}$ and $PM_{2.5}$ concentration. In another research, Lal and Tripathy [47] successfully developed an ANN model for predicting $PM_{10}$ concentration in an open-cast coal mine of India with promising results. Alkasassbeh and Sheta [48] also considered and developed two ANN models based on the Autoregressive with external (ANNARX) method for predicting $PM_{10}$ and total suspended particles (TSP). Their study showed that ANN was a superior technique for predicting $PM_{10}$ and TSP based on several statistical criteria. Mishra, Goyal [49] also developed a hybrid model using ANN and fuzzy logic, i.e., neuro-fuzzy for predicting TSP. As a result, they found that the neuro-fuzzy model provided a performance better than the ANN model. Nagesha, Chandar [50] also conducted a similar study using the ANN model for predicting $PM_{10}$ produced by drilling operations in an open-pit mine. Patra, Gautam [51] also developed an ANN model to predict various particulate matter (i.e., $PM_{0.23-0.3}$, $PM_{0.3-0.4}$, $PM_{0.4-0.5}$, $PM_{0.5-0.65}$, $PM_{0.65-0.8}$, $PM_{0.8-1}$, and $PM_{1-1.6}$) with an excellent result. Table 1 summarizes several AI techniques in predicting $PM_{10}$ concentration and TSP. Also, similar and relevant work on the prediction of $PM_{10}$ concentration can be found at the following references [52–61]. Given the previous work, it can be seen that it is feasible to use AI techniques to predict dust concentration as well as $PM_{10}$ emissions. It has been demonstrated that AI techniques, especially ANN, are significant in predicting and controlling dust concentration, and can form the basis of development of other models and dust emission monitoring methods.

**Table 1.** Summary of works using AI techniques for predicting $PM_{10}$ concentration and TSP.

| No. | Reference | Technique | Objective | Result |
|---|---|---|---|---|
| 1 | Chelani, Gajghate [45] | ANN | $PM_{10}$ | RMSE = 7.9; $R^2$ = 0.89 |
| 2 | McKendry [46] | ANN | $PM_{10}$ | RMSD = 2.21; r = 0.75 |
| 3 | Lal and Tripathy [47] | ANN | $PM_{10}$ | RMSE = 0.0339; d = 0.9969 |
| 4 | Alkasassbeh, Sheta [48] | ANN | $PM_{10}$, TSP | $MSE_{PM10}$ = 219.785; $MSE_{TSP}$ = 1010.7; $MMRE_{PM10}$ = 0.313; $MSE_{TSP}$ = 0.234 |
| 5 | Mishra, Goyal [49] | Neuro-fuzzy | $PM_{2.5}$ | R = 0.72 |
| 6 | Nagesha, Chandar [50] | ANN | $PM_{10}$ | MSE = 0.00606; $R^2$ = 0.96 |
| 7 | Patra, Gautam [51] | ANN | $PM_{0.23-0.3}$ $PM_{0.3-0.4}$ $PM_{0.4-0.5}$ $PM_{0.5-0.65}$ $PM_{0.65-0.8}$ $PM_{0.8-1}$ $PM_{1-1.6}$ | R = 0.806, 0.852, 0.808, 0.896, 0.698, 0.674, 0.788, respectively |

Our literature review showed that, although the studies of $PM_{10}$ concentration prediction in open-pit mines had been conducted, they are, however, still very sketchy. Furthermore, the intensity of $PM_{10}$ concentration in each area/mine/region/country is different due to the characteristics of rock mass as well as the meteorological conditions. Therefore, each area/mine/region needs to be specifically researched in order to have solutions aimed to ensure safety for the health of residents and employees and the surrounding environment. In this study, $PM_{10}$ concentration from drilling operations in open-pit mines was investigated and considered. The Coc Sau open-cast coal mine in Vietnam was selected as a case study for this aim. To this regard, many AI-related techniques, such as deep learning (DL), reinforcement learning (RL), recurrent neural networks (RNN), convolutional neural networks (CNN), generative adversarial networks (GAN), artificial neural network (ANN), and the adaptive-neuro fuzzy inference system (ANFIS), to name a few, can be applied; however, it is challenging to select which technique is the best for analyzing the concentration of $PM_{10}$ in open-pit mines. In practical engineering, simple machine learning and the optimization algorithms

are often selected as the goal of soft computing models for simple regression problems due to their applicability and effectiveness. Therefore, this study considered the feasibility of a simple machine learning algorithm (i.e., Support Vector Regression (SVR)) and an optimization algorithm (i.e., Particle Swarm Optimization (PSO)), for predicting $PM_{10}$ concentration in open-pit mines. A new hybrid model based on an assembly of SVR and PSO, i.e., the PSO-SVR model was developed and proposed in this study for estimating $PM_{10}$ concentration from drilling operations. The obtained results by the proposed PSO-SVR model was compared to and evaluated against the other models, including k-nearest neighbors (KNN), random forest (RF), and classification and regression trees (CART). This study is useful for the environmental science community and practical engineering in minimizing the effect of dust concentration on the surrounding environment.

## 2. Study Area and Geological Conditions

The Coc Sau open-pit coal mine is one of the largest open-pit coal mines in Vietnam, with the depth of −250 m above sea level (MASL). It is located in Cam Pha city, Vietnam, and it lies within latitudes 106°25′00″E–108°05′00″E and longitudes 20°45′30″N–21°10′00″N (Figure 1). The coal production of the mine reached 2 to 3 million ton/year, overburden reached 30 to 40 million m³/year.

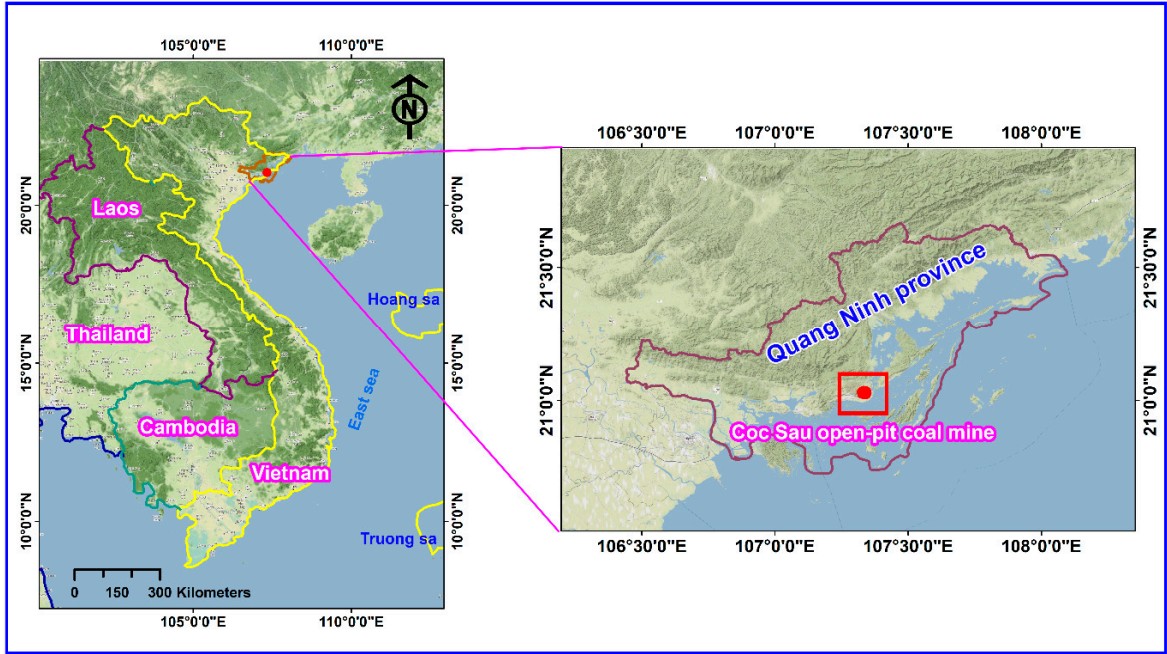

**Figure 1.** Location of the study site.

The Coc Sau open-cast coal mine was covered entirely by sedimentary rocks of the Late Triassic Hon Gai Formation ($T_3$n-rhg) (Figure 2). The formation was composed of gritstone, conglomerate, sandstone, claystone, siltstone, shale, and coal seams. In general, these sedimentary rocks are quite hard with the Protodyakonov strength index (f) of 8 to 11 [62]. Therefore, fragmenting rocks by drilling-blasting was considered as an effective method for the exploitation of coal in the mine.

For drilling operations, СБШ-250, D245S, and DML were the favorite drills used in the mine. The borehole diameters used were in the range of 200 to 250 mm. For drilling, the average drilling speed was 10 to 15 m/h, so the amount of $PM_{10}$ concentration incurred was not small. The drilling contributed to an increase in the impact of dust on the surrounding environment and public health. In this regard, the distance from the mine to the residential area is about 0.42 mile (~700 m); therefore, the effect of dust, especially the $PM_{10}$ concentration, is significant (Figure 3). Also, the occupational hazard for employees is a particular concern.

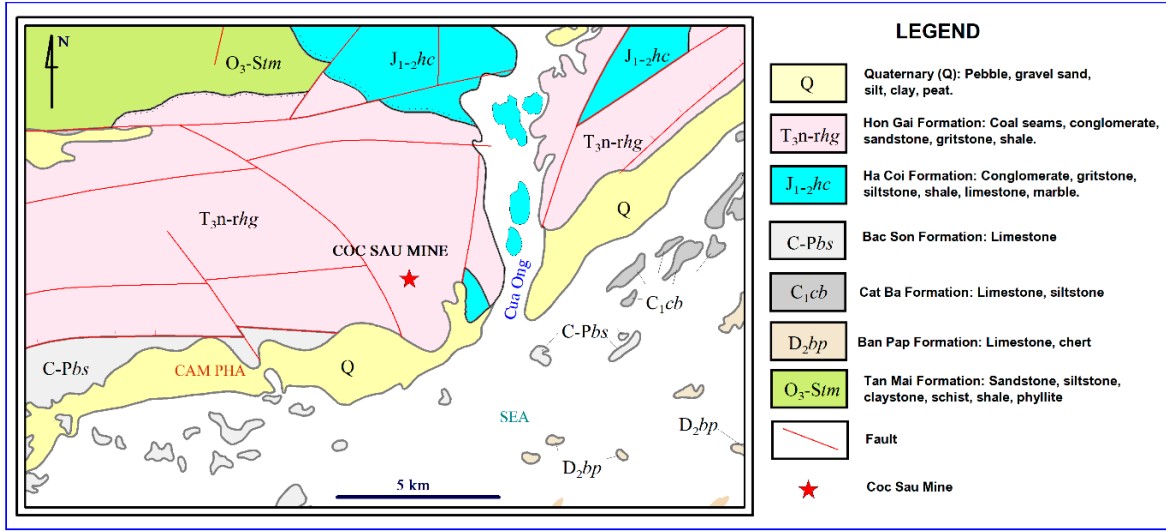

**Figure 2.** Simplified geological conditions of the Coc Sau open-pit coal mine.

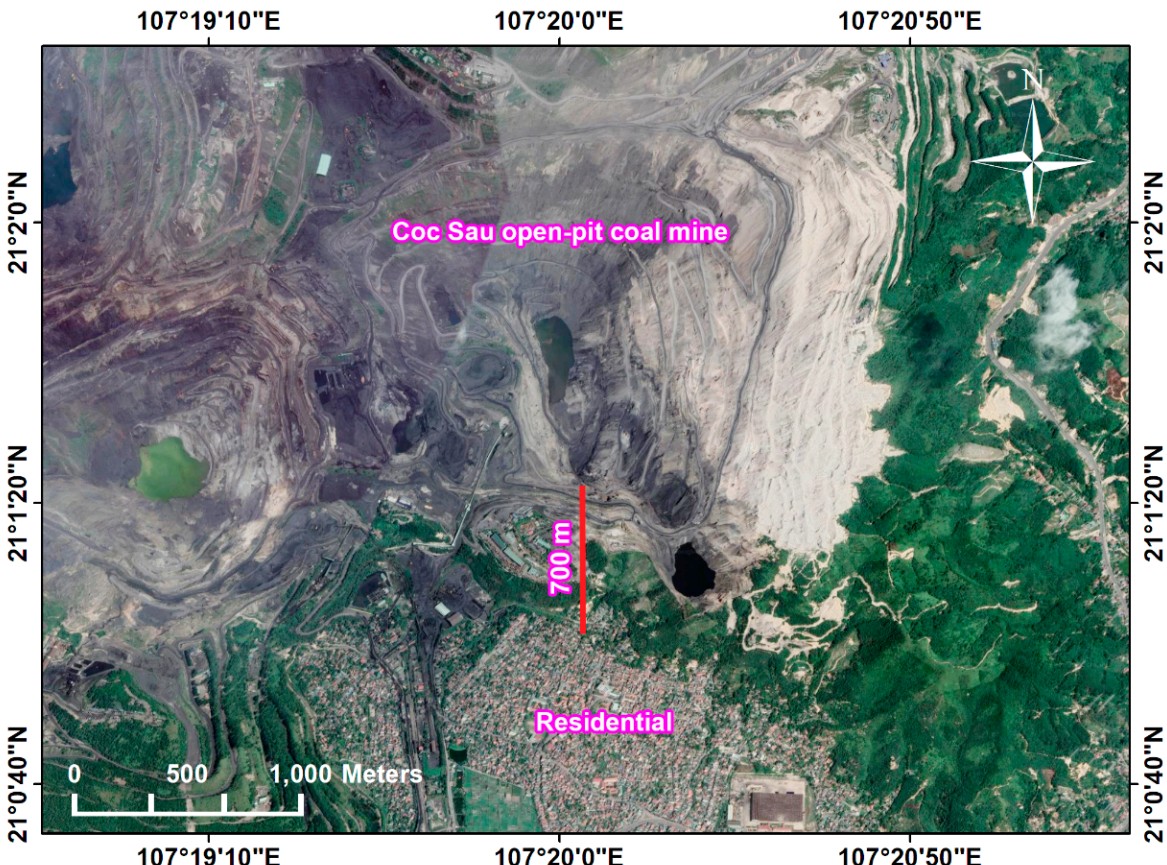

**Figure 3.** The distance from the mine to the residential area.

## 3. Data Collection and Its Characteristics

As stated above, CBIII-250, D245S, and DML drills were used at the mine for drilling boreholes, with the diameter of boreholes in the range of 200 mm to 250 mm. For data collection, the drilling diameter/diameter of boreholes ($d$), the penetration rate of the drill ($P$), the moisture content ($W_{tn}$), the silt content ($S$), the density of rock mass ($\rho$), the compressive strength ($\sigma_c$), and the rebound hardness number ($R$) were used as independent variables to predict $PM_{10}$ concentration.

As mentioned, СБШ-250, D245S, and DML were the component drills used in the mine with $d$ in the range of 200 to 250 mm. With an average depth in the range of 12 to 15 m, the drills can penetrate the rock with $P$ in the range of 0.16 to 0.41 m/min.

For humidity moisture ($W_{tn}$), the core drilling rock samples were carefully preserved and brought to the laboratory. Each drilling hole was collected as a sample by combining representative rock in the drilling core. In the laboratory, a total of 245 samples were used to determine the natural water content of rock ($W_{tn}$). The method was conducted by Vietnam's standard of TCVN 10321:2014 [63]. The equation (1) for calculating the natural water content of the rock is as follows:

$$W_{tn} = \frac{g_1 - g_2}{g_2 - g_0} \cdot 100 \tag{1}$$

where $g_0, g_1, g_2$ are the weights of the sample box without a rock sample, of the sample box and the natural rock sample, and of the sample box and the dried rock sample, respectively.

For silt content ($S$), the fine particles present in the drill cutting was determined (in %). For determining the properties of rock mass ($\rho$, $\sigma_c$), indirect experiments in combination with Protodiakonov's strength index were used. To determine $R$, a rebound hardness tester was used, i.e., a Schmidt hammer.

For measuring $PM_{10}$ concentration, KANOMAX digital dust monitor model 3442 was used in this study (the manual of KANOMAX can be found at the following link https://kanomax.biz/asia/products/dust_monitors/model_3442_3443.html). This instrument (model 3442) is a light scattering portable dust monitor using a semiconductor laser radiation light source [64]. The principle of this method is that when the dust is irradiated, the scattered light intensity from the dust is proportional to the mass concentration. Two monitoring stations with this instrument were installed: One in the upstream and the other in the downstream. To ensure the accuracy of the predictive models, the datasets from independent drilling locations were collected to eliminate the maximum of $PM_{10}$ concentration from other mining operations. In other words, there were no other activities (e.g., loading/unloading, transporting, blasting) on the same work sites of the drilling operation. Furthermore, the dust monitor was set around the drill with the distance being in the range of 5 to 10 m to ensure a maximum of $PM_{10}$ concentration from the drilling operation was obtained (Figure 3). With 245 drilling operations, $PM_{10}$ concentration was measured in the range of 0.148 to 1.306 gm/s. Figure 4 illustrates the data collection process for this study, and the data used is summarized in Table 2. Also, Figure 5 shows the relationship between input and output variables used in this study.

**Table 2.** Characteristics of the data used.

| Elements | $d$ (mm) | $P$ (m/min) | $W_{tn}$ (%) | $S$ (%) |
|---|---|---|---|---|
| Min. | 200.0 | 0.1600 | 0.29 | 15.20 |
| 1st Qua. | 200.0 | 0.2100 | 7.84 | 24.70 |
| Median | 230.0 | 0.2500 | 11.38 | 27.60 |
| Mean | 227.3 | 0.2558 | 11.57 | 27.58 |
| 3rd Qua. | 250.0 | 0.2900 | 15.39 | 30.10 |
| Max. | 250.0 | 0.4100 | 28.12 | 39.20 |
| Elements | $\rho$ (gm/cm$^3$) | $\sigma_c$ (MPa) | $R$ (m) | $PM_{10}$ (gm/s) |
| Min. | 1.220 | 13.00 | 16.00 | 0.148 |
| 1st Qua. | 1.230 | 15.00 | 20.00 | 0.329 |
| Median | 1.240 | 16.00 | 22.00 | 0.474 |
| Mean | 1.243 | 15.98 | 21.63 | 0.496 |
| 3rd Qua. | 1.260 | 17.00 | 23.00 | 0.646 |
| Max. | 1.270 | 19.00 | 27.00 | 1.306 |

Note: drilling diameter/diameter of borehole ($d$), penetration rate of the drill ($P$), moisture content ($W_{tn}$), silt content ($S$), density of rock mass ($\rho$), compressive strength ($\sigma_c$), rebound hardness number ($R$), particulate matter 10 micrometers ($PM_{10}$).

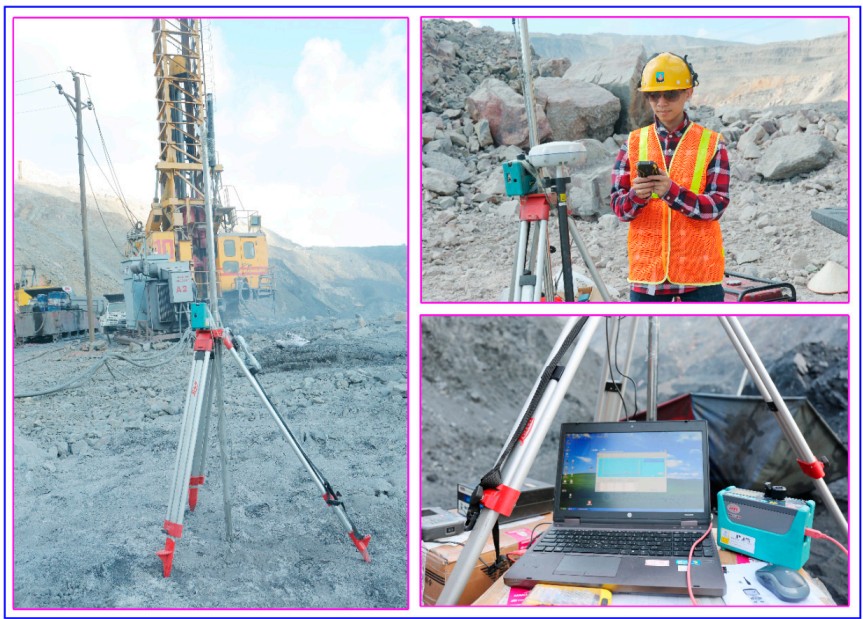

**Figure 4.** Data collection at the Coc Sau open-pit coal mine for this study.

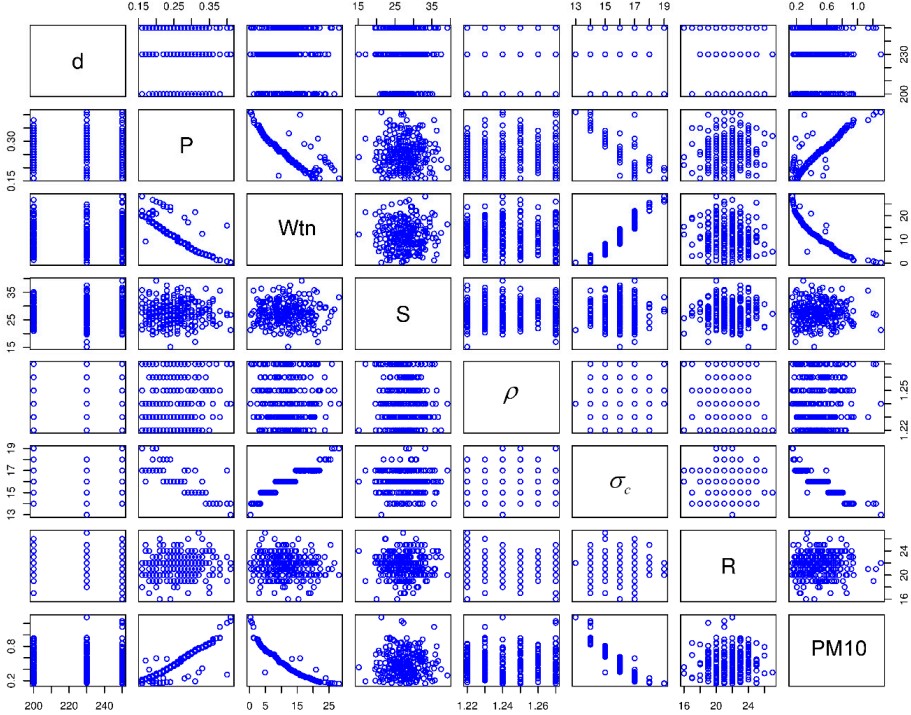

**Figure 5.** Relationship between input and output variables used in this study.

In Figure 5, the spread of the points denotes the relationship between the variables. Each sub-figure means the relationship between two variables. Note that the matrix is symmetrical. The correlated attributes showed good structure in their relationship, such as $\rho$, $W_{tn}$, and $P$. They are not linear, but good predictable curved relationships. Also, Figure 5 shows that the correlation of most of the input variables was not high. It is a right candidate for AI techniques in finding out the association between the input and output variables.

## 4. Compositions Analysis of $PM_{10}$ from Drilling Operations

To assess the danger level of $PM_{10}$ from drilling operations to community health and surroundings, the compositions of $PM_{10}$ from drilling operations at the Coc Sau open-pit coal mine were analyzed by energy dispersive X-ray spectroscopy (SEM-EDS) (Quanta 450—FEI Company, Hillsboro, OR, USA). Accordingly, three dust samples of different boreholes were collected at the Coc Sau open-pit coal mine for analysis of their size and composition. The results from the scanning electron microscope coupled with SEM-EDS showed that the dust particles from the drilling process in the study area were relatively small with a size of less than ten μm. The composition of the dust consisted of many different elements, such as Al, Si, K, Fe (Figure 6), which are very dangerous for human health when breathed in. Therefore, accurate predictions of $PM_{10}$ concentration from drilling operations in open-pit mines are necessary to minimize impact on public health and the surrounding environment.

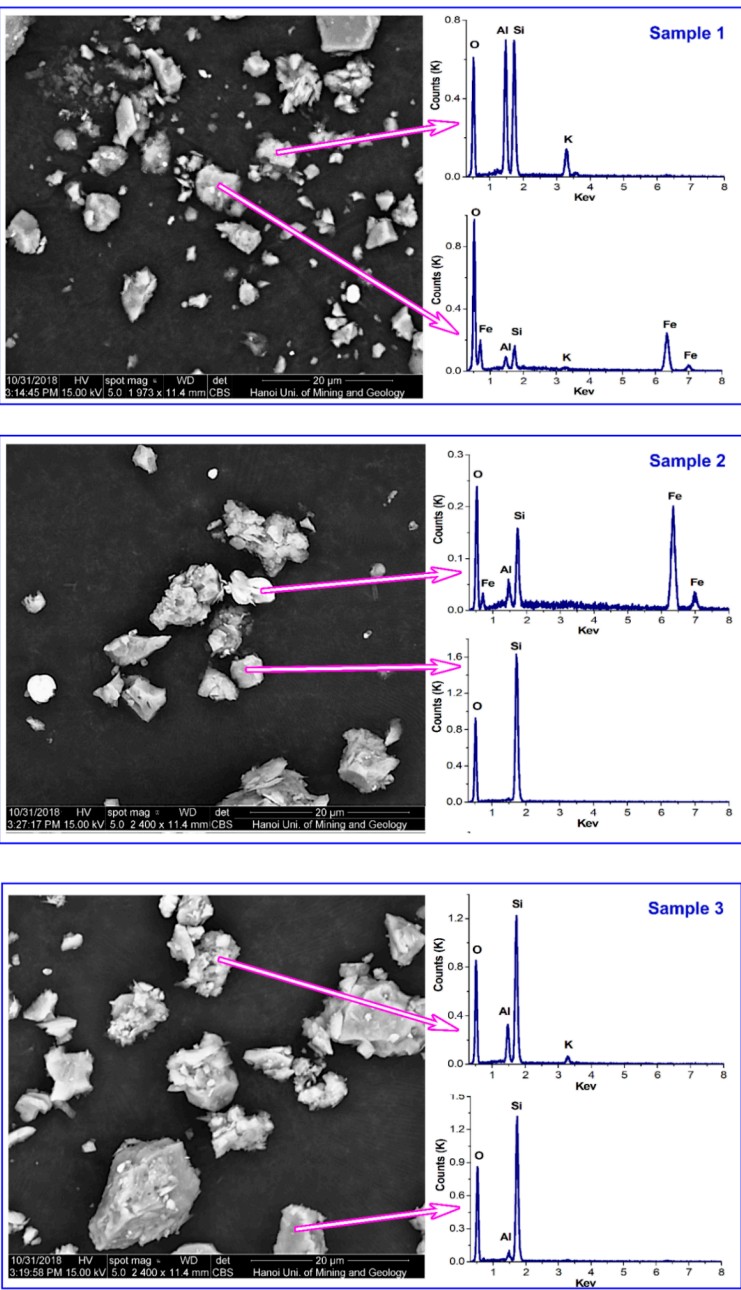

**Figure 6.** The composition of dust particles in drilling operations at the Coc Sau open-pit coal mine. The *y*-axis depicts the number of counts, and the *x*-axis represents the energy of the X-rays.

### 5. The Principle of Intelligence Techniques

*5.1. Random Forest (RF)*

In this section, the RF is briefly presented. It is well-known as a decision tree algorithm, which was proposed by Breiman [65]. RF is a classification or regression technique using classification or regression trees in a group as a forest. Inspired by the development of the forest, each tree in the forest is treated as a decision. For regression problems (such as estimating $PM_{10}$ concentration from drilling operations in this study), the average value of voters is selected as the final decision of the RF model as well. In other hands, RF is considered as an ensemble learning method based on the propensity of each tree [66]. It uses the bagging technique and decision tree algorithm like the CART algorithm to build an extensive collection of the correlate-decision trees. In statistics, RF uses the bagging technique as an ensemble tool to improve the accuracy and stability of models. The bootstrap technique with replacement is also used by RF to generate decision trees without prune [67]. For each bootstrap sample, a model is developed as a predictor. An ensemble of predictors and its average value is computed for the final decision of the RF (for regression problems) as follows (Equation (2)):

$$\hat{f}(x) \; = \; \frac{1}{N} \sum_{n\,=\,1}^{N} \hat{f}_n(x) \tag{2}$$

where $N$ denotes the number of bootstrap samples; $b$ denotes predictor; $\hat{f}(x)$ denotes the prediction of the $n$ model on an observation $x$.

A literature review showed that the RF algorithm has never been used to estimate $PM_{10}$ concentration from drilling operations; for this study, it was considered and developed for estimating $PM_{10}$ concentration from drilling operations. The background of the RF algorithm can be further found in [68–71].

*5.2. Support Vector Regression (SVR)*

SVR is one of the branches of the support vector machine (SVM) algorithm, one of the benchmark algorithms used in machine learning first introduced by Vapnik [72]. It has two main categories, including SVR and SVC (support vector classification). Of those, SVR is considered as the most common application form of continuous problems. The details of the SVR is presented in reference [73]. Also, kernel functions can be applied for regression problems (Equations (3)–(6)), including:

Linear function:
$$T(a,b) \; = \; a \cdot b \tag{3}$$

Polynomial function:
$$T(a,b) \; = \; [(a \cdot b) + 1]^d ; d \in (1,n) \tag{4}$$

Radial original kernel function:
$$T(a,b) \; = \; \exp[\frac{-\|a - b\|^2}{\sigma^2}] \tag{5}$$

Two-layer neural function:
$$T(a,b) \; = \; \tanh[e(a \cdot b) - \delta] \tag{6}$$

Like the RF algorithm, a review of previous works showed that the SVR algorithm has never been used to estimate $PM_{10}$ concentration from drilling operations. Therefore, it was considered and combined with the PSO algorithm for estimating $PM_{10}$ concentration from drilling operations in this study. More principles of the SVR algorithm can be seen in [72–77].

### 5.3. Classification and Regression Tree (CART)

CART is a statistical technique proposed by Breiman [78]. It can solve both regression and classification problems for exploring as well as modeling data. CART was also developed based on decision trees to explain the variation of the dependent variable by one or more independent variables [79]. As a "white box" algorithm, the relationship between dependent and independent variables is more straightforward with the CART algorithm [80,81]. It does not consider any previous assumptions related to the relationship between variables. Instead, the repeatedly split data process is applied to construct the tree based on an independent variable. The data is separated into two reciprocally exclusive clusters at each split, each of which is as much the same as possible. Then, the separation process is held to each group independently. Yes/no answers concerning the predictor values are the primary basis for generating a binary tree by the CART algorithm.

For developing a CART model for predicting $PM_{10}$ concentration, the four main steps of the CART algorithm are conducted as below:

1. Applying some rules to exploit data at a node based on a variable value;
2. Using some criteria to prevent the creation of complex trees;
3. Pruning for optimum performance of the model;
4. Calculation and prediction of the output for terminal nodes.

As with the RF and SVR algorithms, a review of previous studies show that the CART algorithm has never been used to estimate $PM_{10}$ concentration from drilling operations. Therefore, it is considered and developed in this study. More background of the CART algorithm can be referred to at [82–86].

### 5.4. K-Nearest Neighbors (KNN)

KNN is one of the "learning lazy" algorithms, which can be used/applied for regression as well as classification problems [87]. KNN aims to find the number of nearest neighbors ($k$) from a functional space. To obtain this goal, the Euclidean distance $\|\vec{x} - \vec{y}\|$ between the input variables and queries is calculated as Equation (7). Then, the $k$ closest input points for the consultation is determined.

$$d(a_t, a_i) = \sqrt{\sum_{n=1}^{N} W_n (a_{t,n} - a_{i,n})^2} \tag{7}$$

where $N$ is the features number; $\alpha_{i,n}$ is the $n$th feature value of the training point $\alpha_i$; $\alpha_{t,n}$; is the $n$th feature value of the testing point $\alpha_t$; $W_n$ is the weight of the $n$th feature, $0 \leq W_n \leq 1$.

For estimating a regression problem (such as predicting $PM_{10}$ concentration in this study), a kernel function is often applied to calculate $W_n$ based on its proximity to the testing point (Equation (8)), as follows:

$$\hat{f}(a_t) = \frac{\sum_{i=1}^{k} \phi(a_t, a_{(i)}) f(a_{(i)})}{\sum_{i=1}^{k} \phi(a_t, a_{(i)})} \tag{8}$$

Where $\phi(a_t, a_{(i)})$ denotes a kernel function at the $i$th training point $\alpha_i$; $f(a_{(i)})$ indicates the response of $\alpha_i$.

Like the RF, SVR, and CART algorithms, a review of the literature showed that the KNN algorithm has never been used to estimate $PM_{10}$ concentration from drilling operations, and is considered and developed to predict $PM_{10}$ concentration from drilling operations in this study. More principles of KNN algorithm can be seen in [42,87–89].

*5.5. Particle Swarm Optimization (PSO) Algorithm*

PSO is a swarm algorithm inspired by the behavior of the particles/social animals, such as fish or birds. It was introduced and developed by Eberhart and Kennedy [90] and classified as one of the metaheuristic techniques. It was considered as an evolutionary computation technique in the statistical community [41]. The PSO algorithm implements six steps for optimal searching as the following procedure. In which, the velocity and local positions are continuously updated using Equations (9) and (10), for sharing their experience.

1. Generate the particle swarm size with random locations and speeds on $d$ dimensions in the subject space.
2. Estimate the proper optimization fitness function for each particle in $d$ factors.
3. Compare the fitness evaluation with *pbest* of the particle. If the present value is better than *pbest*, then set *pbest* value equal to the present value, and the *pbest* position equal to the current position in $d$-dimensional space.
4. Compare the evaluation of the fitness with the overall previous best of the population. If the current value is better than *gbest*, then reset *gbest* to the current population index and value of particle.
5. Change the velocity and local of the particle as Equations (9) and (10) as follow:

$$v_j^{i+1} = wv_j^{(i)} + (c_1 \times r_1 \times (local\ best_j - x_j^{(i)})) + (c_2 \times r_2 \times (global\ best_j - x_j^{(i)})),$$
$$v_{\min} \leq v_j^{(i)} \leq v_{\max} \tag{9}$$

$$x_j^{i+1} = x_j^{(i)} + v_j^{(i+1)}; j = 1, 2, \ldots, n \tag{10}$$

where $x_j^{(i)}$ denote the position of particle $j$ at iteration $i$; $v_j^{(i)}$ denote the velocity of particle $j$ at iteration $i$; $w$ is the inertial weight coefficient; $i$ is the iteration number; $r_1$ and $r_2$ represent a random number uniformly distributed in [0,1] which is randomly generated at iteration and for each particle.

6. Loop to step (2) until a criterion is met. Usually, the maximum number of iterations or sufficiently good fitness, the searching process will stop.

In this study, the PSO algorithm was used/applied to optimize the hyper-parameters of the SVR model with different kernel functions (i.e., radial basis function, polynomial, linear) for predicting $PM_{10}$ concentration from drilling operations.

*5.6. Development of the $PM_{10}$ Concentration Predictive Models*

To establish the $PM_{10}$ concentration predictive models, a database including 245 drilling operations was randomly divided into two phases. In the first phase, ~80% of the data (approximately 221 observations) was used for the development of the models. The remaining ~20% (around 24 observations) in the second phase was used as new data for validating the performance of the developed models. ArcGIS software was used to manage databases with big data query capabilities [91]. The preparation of data for this study is illustrated in Figure 7.

To validate/evaluate the models performances, two performance indicators, the root-mean-squared error (RMSE) and the determination coefficient ($R^2$), were calculated (Equations (11) and (12)):

$$RMSE = \sqrt{\frac{1}{n} \sum_{i=1}^{n} (y_i - \hat{y}_i)^2} \tag{11}$$

$$R^2 = 1 - \frac{\sum_i (y_i - \hat{y}_i)^2}{\sum_i (y_i - \overline{y})^2} \tag{12}$$

where *n* is a total number of data; $y_i$, $\hat{y}_i$ and $\overline{y}$ are measured, predicted, and mean of $y_i$ values, respectively. For an optimal model, $R^2 = 1$ and RMSE = 0.

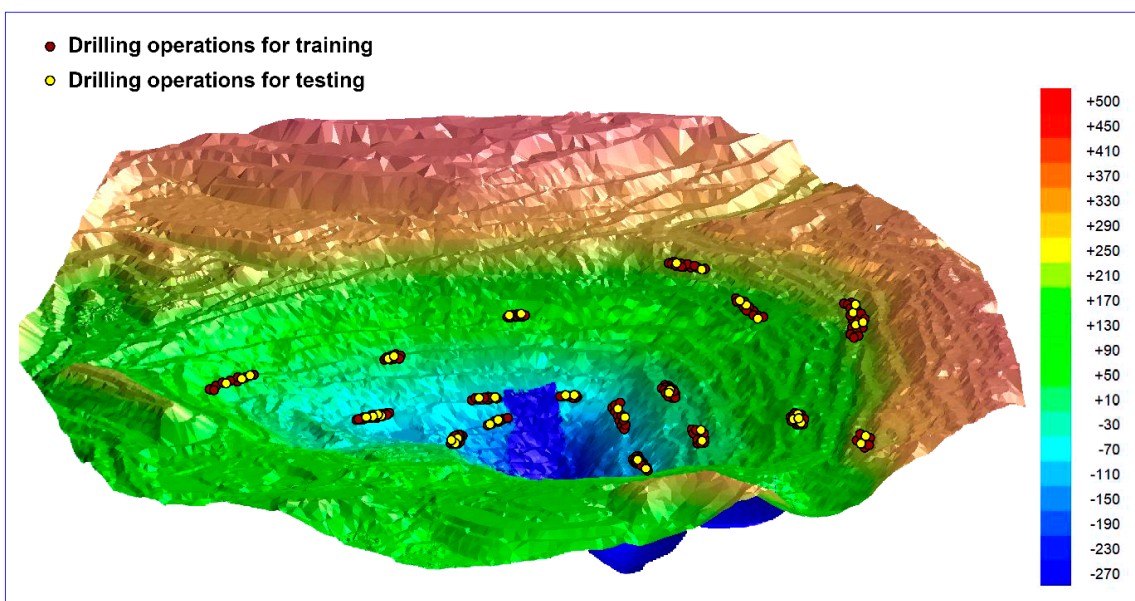

**Figure 7.** The database used for predicting drill-induced $PM_{10}$ concentration in this study.

*5.7. RF Model*

For the RF model, the number of decision trees (*ntree*) and the random predictor selected (*mtry*) are considered as the essential criteria for measuring the quality of the model. To ensure the diversity of the forest, *ntree* was selected equal to 2000 [71] in this study. Each decision tree plays a role as a voter in the RF model. The remaining parameter, i.e., *mtry* was selected by a grid search technique with *mtry* in the range of 1 to 50 (Figure 8). Also, *k*-fold cross-validation resampling technique was applied with *k* = 10 to avoid overfitting/underfitting. As a result, the optimal values of the RF model for anticipating $PM_{10}$ concentration was obtained at *ntree* = 2000 and *mtry* = 45.

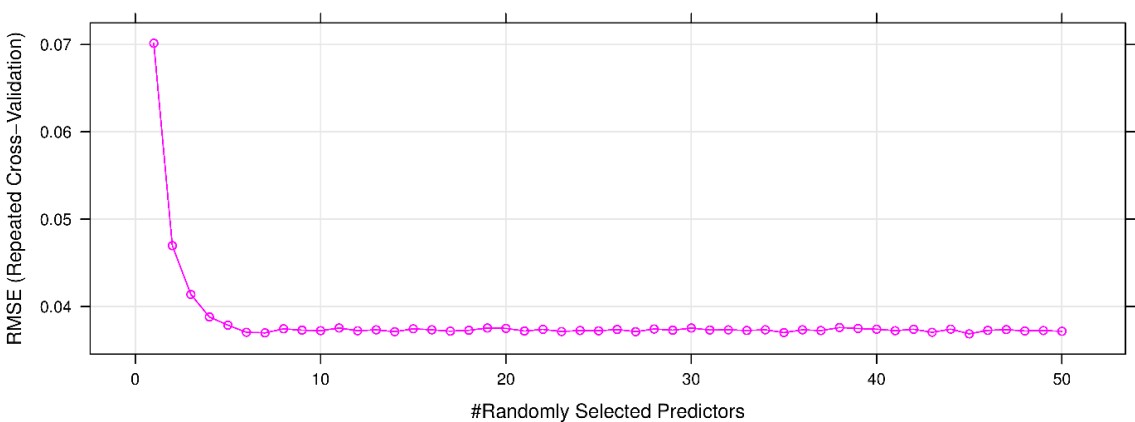

**Figure 8.** Performance of the random forest (RF) model for forecasting $PM_{10}$ concentration on the training dataset.

### 5.8. CART Model

For the CART model, the complexity parameter (*cp*) was used to tune the performance of the model. Similar to the RF model, a grid search with the *cp* in the range of 0 to 0.1 was conducted. The 10-fold cross-validation resampling technique was also used to avoid overfitting/underfitting, as those used for the RF model. Eventually, the CART model obtained the best performance at *cp* = 0 for estimating $PM_{10}$ concentration, as shown in Figure 9.

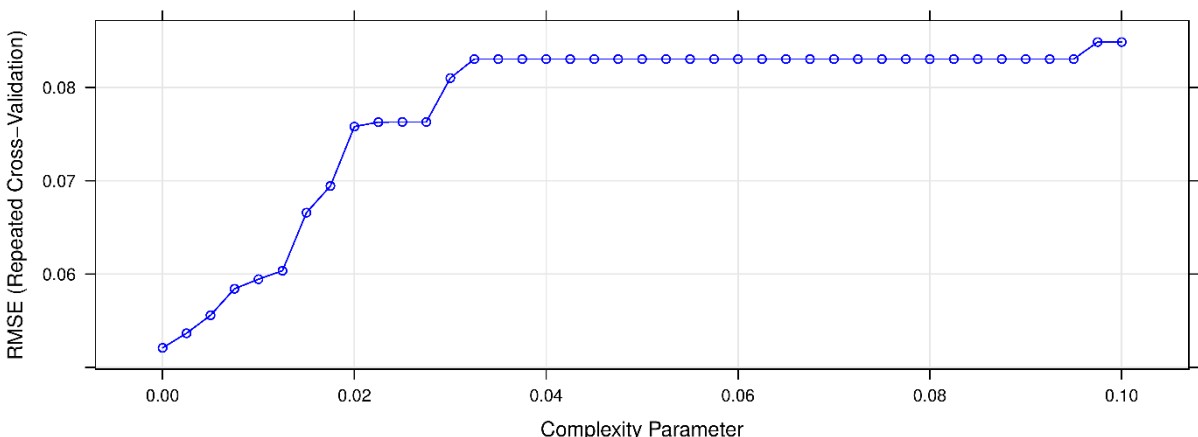

**Figure 9.** Root-mean-squared error (RMSE) of the classification and regression trees (CART) model with various *cp* values on the training dataset.

### 5.9. KNN Model

For the KNN model, its performance is controlled by the number of neighbors (*k*). Similar to the RF and CART models, a grid search with the *k* in the range of 1 to 50 was conducted to find out the best KNN model. The 10-fold cross-validation technique with three repeats was also used to avoid overfitting/underfitting of the KNN model. Finally, a good KNN model was found at *k* = 4, as shown in Figure 10.

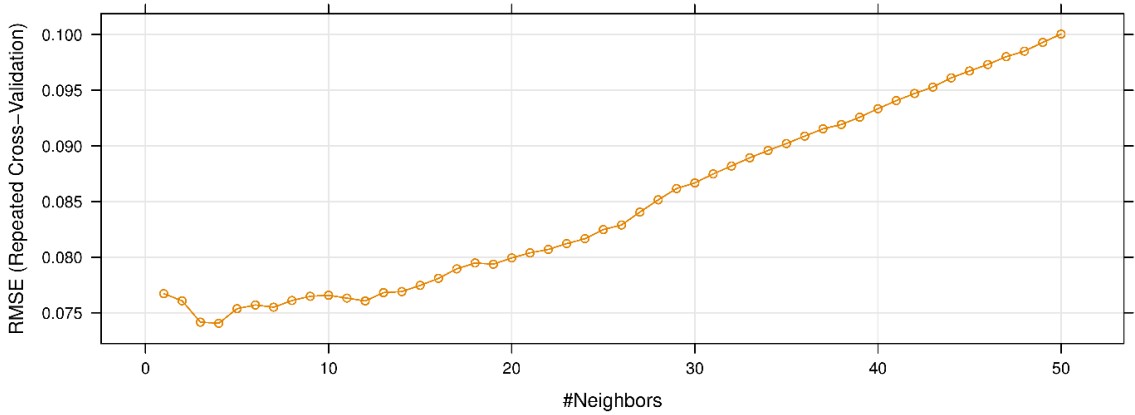

**Figure 10.** Performance of the k-nearest neighbors (KNN) model with various *k*.

### 5.10. PSO-SVR Models for Estimating $PM_{10}$ Concentration

As introduced above, the primary purpose of this study is developing and proposing a new artificial intelligence system for predicting $PM_{10}$ concentration, i.e., PSO-SVR model. Three forms of kernel function were applied for the development of the PSO-SVR models, abbreviated as PSO-SVR-L, PSO-SVR-P, and PSO-SVR-RBF models. In this procedure, the PSO algorithm performs a global search for the optimal values of SVR models through a fitness function, i.e., RMSE (Equation 11). For each PSO-SVR model, the hyper-parameters are different. Depending on the kernel function used,

the hyper-parameters of each PSO-SVR model are different. Table 3 shows the hyper-parameters of the PSO-SVR models with different functions of the kernel. A framework of the PSO-SVR model for estimating concentration in this study is shown in Figure 11.

**Table 3.** Hyper-parameters of the SVR models with the different kernel functions.

| Model | Hyper-Parameters | | | |
|---|---|---|---|---|
| | $C$ | $d$ | $\gamma$ | $\Sigma$ |
| PSO-SVR-L | ✓ | - | - | - |
| PSO-SVR-P | ✓ | ✓ | ✓ | - |
| PSO-SVR-RBF | ✓ | - | - | ✓ |

Note: cost ($C$); degree ($d$); scale ($\gamma$); sigma ($\sigma$).

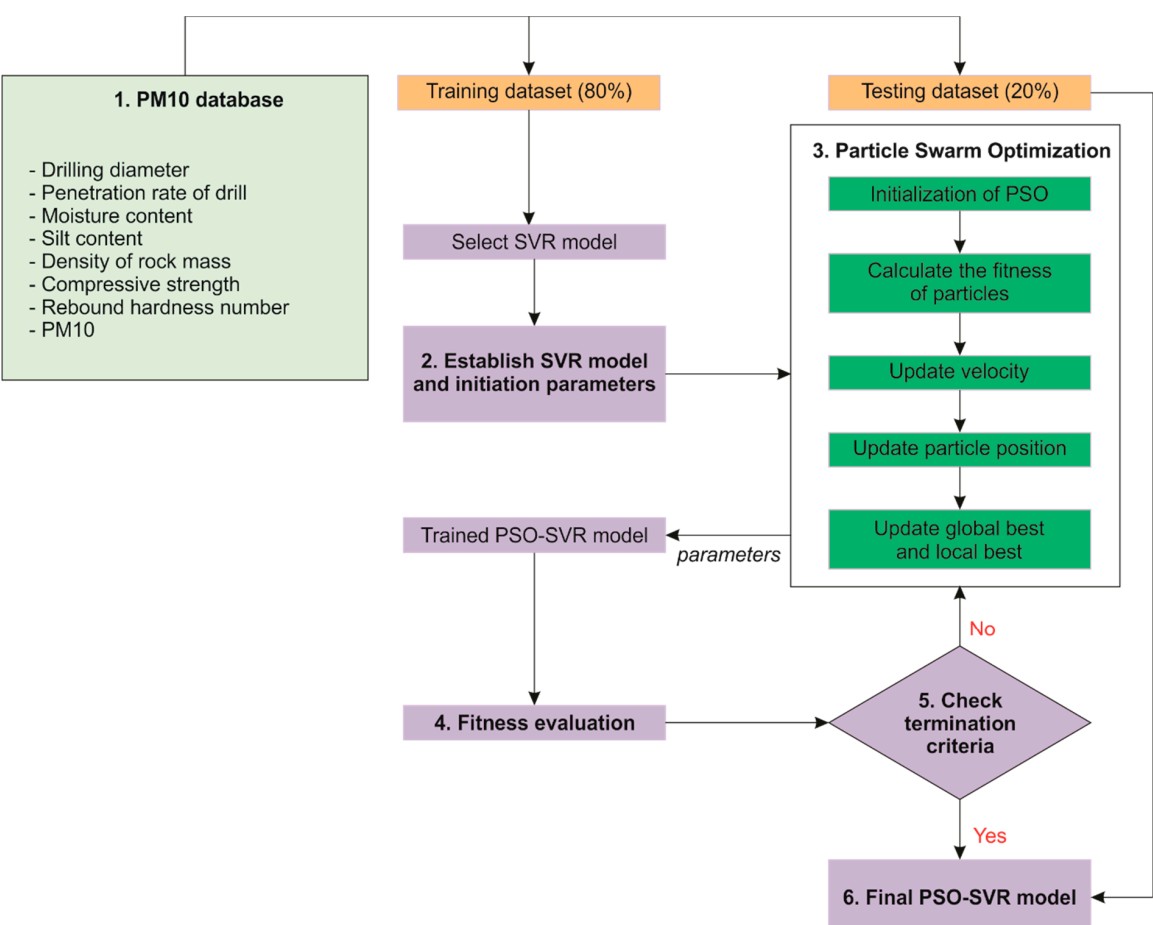

**Figure 11.** Proposing the PSO-SVR model for estimating PM$_{10}$ concentration.

Before proceeding to search for the optimal values of hyper-parameters of the SVR models, the parameters of the PSO algorithm is set as the first step on the training dataset. In the PSO algorithm, the number population ($p$), maximum number of iteration ($m_i$), maximum particle's velocity ($V_{max}$), individual cognitive ($\phi_1$), group cognitive ($\phi_2$), and inertia weight ($w$) are the parameters used for the optimization process. The sample size should have functional population diversity [92–94], therefore, a trial-and-error procedure was employed with the swarm size was 100, 150, 200, 250, 300, respectively ($p$ = 50, 100, 150, 200, 250, 300). For terminating the optimization process in this study, $m_i$ was set equal to 1000 to check the fitness of particle positions using the RMSE metric (Equation 11). To ensure the balance between global discovery and local search, $w$ was set equal to 0.9 [95]. According to

Kennedy [96] and Clerc and Kennedy [97], $\phi_1$ should be equal to $\phi_2$ and $\phi_1 + \phi_2$ lie in the range of 0 to 4. Therefore, in this study, $\phi_1 = \phi_2 = 1.6$. To ensure convergence and prevent explosion [98], $V_{max}$ was set equal to 2 in this study. Figures 12–14 shows the training process of PSO-SVR models (PSO-SVR-L; PSO-SVR-P; PSO-SVR-RBF).

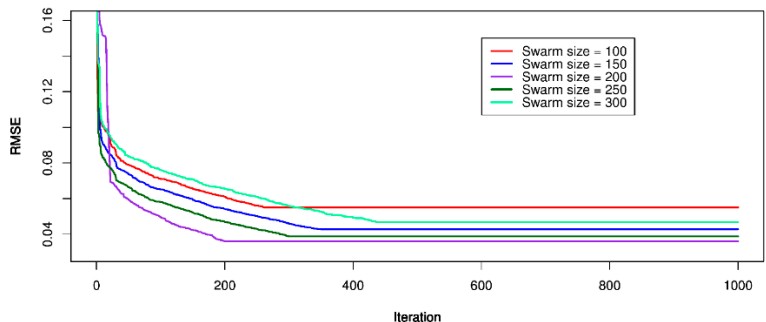

**Figure 12.** Performance of the PSO-SVR-L model on the training dataset.

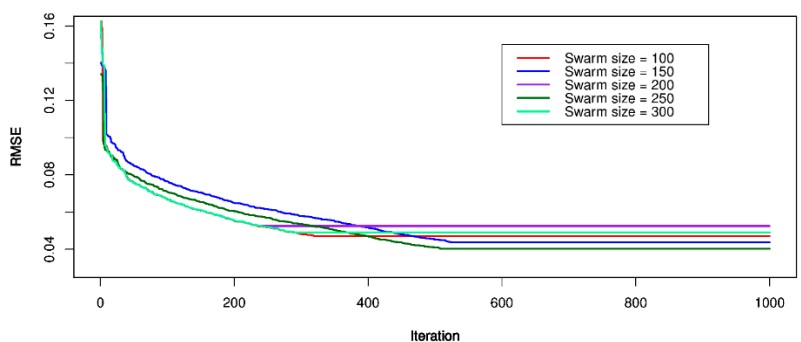

**Figure 13.** Performance of the PSO-SVR-P model on the training dataset.

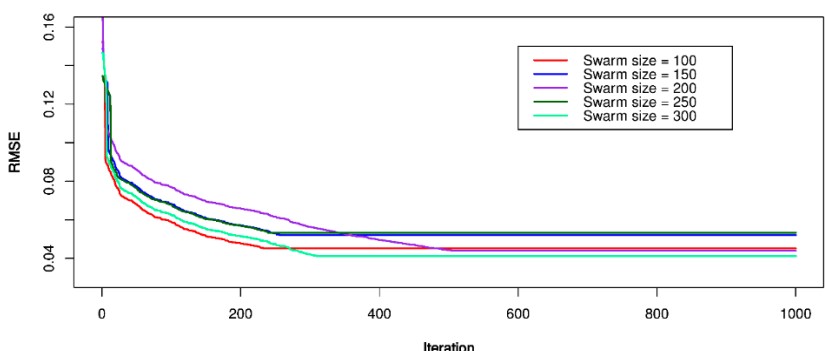

**Figure 14.** Performance of the PSO-SVR-RBF model on the training dataset.

After the optimization process of the PSO algorithm was stopped, the optimal values of the SVR models were extracted for predicting $PM_{10}$ concentration. Table 4 shows the obtained results by the PSO algorithm for searching the hyper-parameters of the SVR models.

**Table 4.** The parameters of the PSO-SVR models for estimating $PM_{10}$ concentration.

| Model | Hyper-Parameters | | | |
|---|---|---|---|---|
| | $C$ | $d$ | $\gamma$ | $\sigma$ |
| PSO-SVR-L | 900.792 | - | - | - |
| PSO-SVR-P | 509.611 | 2 | 0.0014 | - |
| PSO-SVR-RBF | 19.988 | - | - | 0.01 |

## 6. Results and Discussion

Once the $PM_{10}$ concentration predictive models were developed, their performance was evaluated and discussed through RMSE and $R^2$, as mentioned in Equations (11) and (12). Table 5 shows the performance of the developed models on the training and testing datasets.

**Table 5.** Performance indices of the $PM_{10}$ concentration predictive models.

| Model | Training | | Testing | |
|---|---|---|---|---|
| | **RMSE** | **$R^2$** | **RMSE** | **$R^2$** |
| RF | 0.057 | 0.963 | 0.060 | 0.894 |
| CART | 0.052 | 0.945 | 0.052 | 0.924 |
| KNN | 0.074 | 0.904 | 0.067 | 0.867 |
| **PSO-SVR-L** | **0.036** | **0.963** | **0.040** | **0.954** |
| PSO-SVR-P | 0.040 | 0.962 | 0.042 | 0.948 |
| PSO-SVR-RBF | 0.041 | 0.962 | 0.043 | 0.946 |

Note: The bold type represents the best model. RMSE denotes root-mean-square error; $R^2$ denotes determination coefficient.

Table 5 showed that the PSO-SVR models performed very well in estimating $PM_{10}$ concentration from drilling operations in this study with an RMSE in the range of 0.036 to 0.041 on the training dataset, 0.040 to 0.043 on the testing dataset; $R^2$ in the range of 0.962 to 0.963 on the training dataset, 0.946 to 0.954 on the testing dataset. Notable, the PSO-SVR-L model yielded the best performance with an RMSE of 0.036 and 0.040 for the training and testing dataset, respectively; and $R^2$ of 0.963 and 0.954 for the training and testing dataset, respectively. With the highest performance, the PSO-SVR-L model is the most dominant model for estimating $PM_{10}$ concentration in this study. The remaining AI models (RF, CART, KNN) provided more mediocre performance than those of the PSO-SVR models with an RMSE in the range of 0.057 to 0.074 on the training dataset, 0.052 to 0.067 on the testing dataset; and $R^2$ in the range of 0.904 to 0.963 on the training dataset, 0.867 to 0.924 on the testing dataset. Remarkable, the KNN model yielded the most inferior performance with an RMSE of 0.074 and 0.067 for the training and testing dataset, respectively; and $R^2$ of 0.904 and 0.867 for the training and testing dataset, respectively. Figure 15 shows the measured versus predicted values of $PM_{10}$ concentration on the testing dataset.

Based on the obtained results from the developed models, it can be seen that machine learning algorithms perform very well in estimating $PM_{10}$ concentration from drilling operations in open-pit mines. In particular, the PSO algorithm played a significant role in optimizing the hyper-parameters of the SVR models to provide more accurate predictive results. The PSO-SVR-L model was introduced as the most superior model in this study for estimating $PM_{10}$ concentration that explained the linear relationship of the input variables with $PM_{10}$ concentration. However, the present study used seven independent/input variables for predicting $PM_{10}$, and the effectiveness of the models was different. Therefore, an importance analysis procedure of the independent/input variables for each model was performed in this study, i.e., using the Hilbert-Schmidt Independence Criterion (HSIC) [99,100]. An overall picture of performance as well as the ability to explain the relationship between the independent/input variables of the models is illustrated in Table 6.

From Table 6, it is easy to recognize that $P$, $W_{tn}$, and $\sigma_c$ are the most importance parameters of all developed AI models. They should be used for estimating $PM_{10}$ concentration from drilling operations. Taking a closer look at Table 6 shows that the assessment of the importance of input variables has a significant effect on the performance of the model. In the proposed PSO-SVR-L model, the $W_{tn}$ parameter was most importance parameter with indices of 0.975, whereas the remaining models were only evaluated in the range of 0.896 to 0.957. Likewise, $P$ and $\sigma_c$ also received different values from different models. It can be seen that the evaluation of the importance of input variables as too low or too high has a significant impact on the performance of the forecasting model.

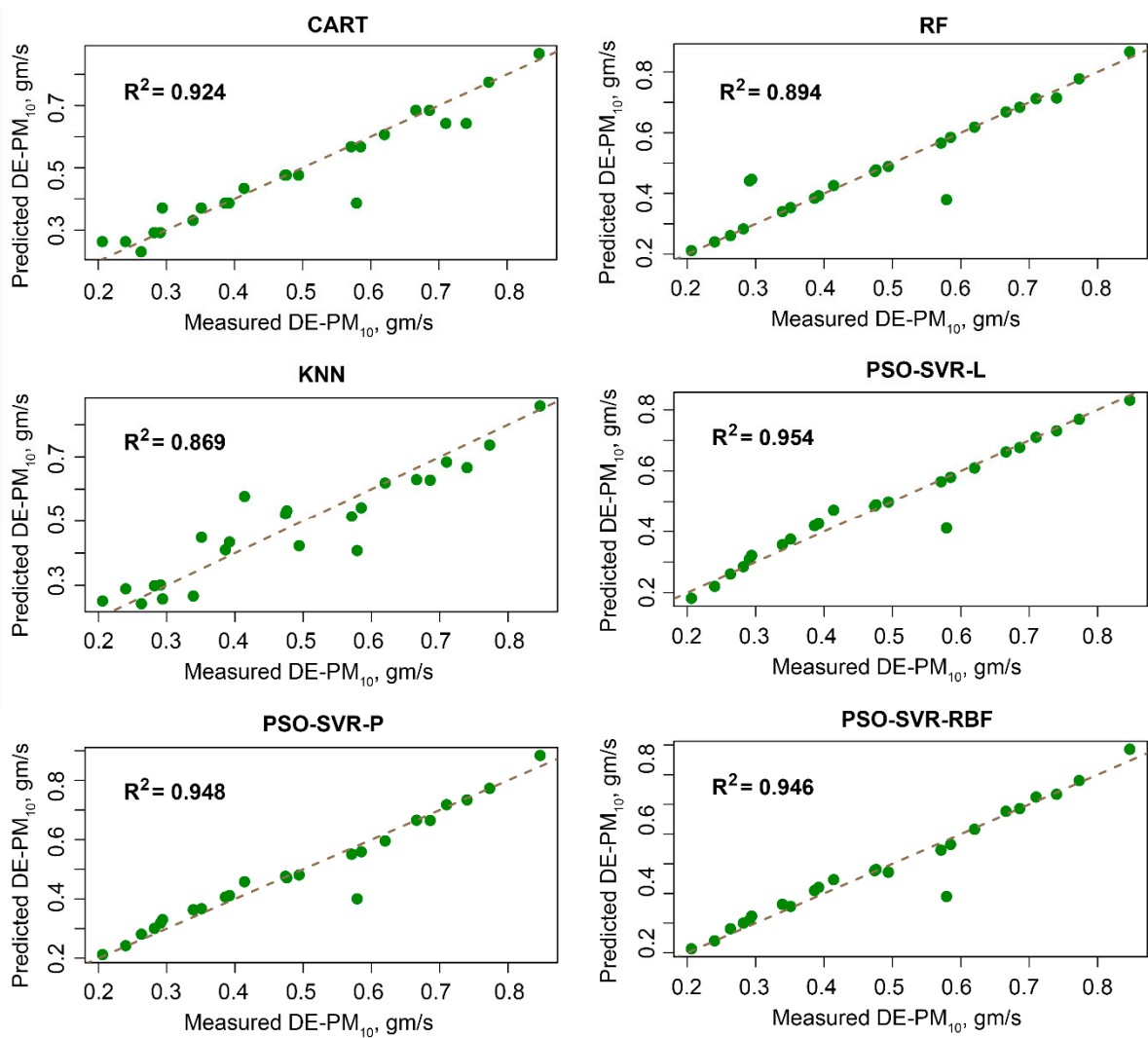

**Figure 15.** Measured versus predicted values of $PM_{10}$ concentration on the testing dataset.

**Table 6.** Important parameters of the models in this study.

| Model | Performance | | | Importance Parameters | | | | | |
|---|---|---|---|---|---|---|---|---|---|
| | **RMSE** | **$R^2$** | **$d$** | **$P$** | **$W_{tn}$** | **$S$** | **$\rho$** | **$\sigma_c$** | **$R$** |
| RF | 0.060 | 0.894 | 0.018 | 0.825 | 0.927 | 0.008 | 0.017 | 0.779 | 0.013 |
| CART | 0.052 | 0.924 | 0.019 | 0.824 | 0.929 | 0.007 | 0.017 | 0.784 | 0.012 |
| KNN | 0.067 | 0.867 | 0.024 | 0.802 | 0.896 | 0.006 | 0.022 | 0.826 | 0.014 |
| PSO-SVR-L | 0.040 | 0.954 | 0.018 | 0.825 | 0.975 | 0.006 | 0.018 | 0.817 | 0.012 |
| PSO-SVR-P | 0.042 | 0.948 | 0.017 | 0.836 | 0.957 | 0.007 | 0.018 | 0.812 | 0.013 |
| PSO-SVR-RBF | 0.043 | 0.946 | 0.016 | 0.839 | 0.955 | 0.006 | 0.018 | 0.813 | 0.013 |

Note: drilling diameter/diameter of borehole ($d$), penetration rate of the drill ($P$), moisture content ($W_{tn}$), silt content ($S$), density of rock mass ($\rho$), compressive strength ($\sigma_c$), rebound hardness number ($R$), particulate matter 10 micrometers ($PM_{10}$), root-mean-squared error (RMSE), determination coefficient ($R^2$).

## 7. Conclusions and Recommendations

Drilling-blasting is an essential task in mining operations. Until now, it is still the most efficient method for fragmenting rocks in open-cast mines. However, the impact on the environment caused by drilling operations as well as blasting operations is inevitable, especially in dust concentration. Evidence of the dangers of $PM_{10}$ has been found in open-pit mines. Therefore, the prediction and strict control of $PM_{10}$ concentrations is necessary to minimize the harmful impact on the surrounding

environment. In this study, $PM_{10}$ concentration from drilling operations in open-pit mines was considered and evaluated. In this study, some conclusions and remarks were drawn:

1.  The dust concentration caused by activities in open-pit mines is high and very dangerous. The dust is the cause of the harmful impact on the environment and public health. In this study, $PM_{10}$ concentration from drilling operations in open-pit mines was considered and predicted. However, the dust concentration caused by other activities in open-pit mines also needs to be controlled and predicted in the future. Total dust concentration needs to be managed in open-pit mines, and is a challenge for future works.

2.  For drilling operations, AI techniques are the advanced methods for predicting drill-induced dust concentration. They provided predictive intelligence models with high accuracy in practical engineering. In addition to the ANN, which was developed by previous researchers, the other machine learning techniques, i.e., RF, CART, KNN, PSO-SVR in this study were also techniques to control air quality in open-pit mines.

3.  The PSO algorithm is a robust tool for the optimization of the SVR model for estimating $PM_{10}$ concentration. With an RMSE of 0.040 and $R^2$ of 0.954, the proposed PSO-SVR-L model was the most dominant model for predicting drill-induced $PM_{10}$ concentration in this study. It should be applied in practical engineering to control $PM_{10}$ concentration from drilling operations as well as the other processes. Additionally, the CART, RF, and KNN models should also be considered in other conditions of different sites for predicting dust concentration. They can be good models in other case studies for predicting the environmental issues in open-pit mines.

4.  $\rho$, $W_{tn}$, and P are the most influential parameters on the $PM_{10}$ concentration predictive model, especially $W_{tn}$. They should be of particular interest and carefully collected for predicting drill-induced $PM_{10}$ concentration.

5.  Based on the obtained results of this study, $PM_{10}$ concentration from drilling operations can be predicted and controlled by the proposed PSO-SVR model. However, there are several operations, such as blasting, transporting, loading/unloading, which are also the causes of dust generation in open-pit mines. Therefore, the feasibility of AI techniques is also needed to investigate and establish a comprehensive air quality control system in open-pit mines.

**Author Contributions:** Data collection and experimental works: H.-B.B., N.Q.L., Q.-T.L., V.-D.N., N.-B.N.; Writing, discussion, analysis: X.-N.B., C.W.L., H.N., H.M.

**Funding:** This research received no external funding.

**Acknowledgments:** This work was financially supported by the Ministry of Education and Training (MOET) in Viet Nam under grant number B2018-MDA-03SP. The authors also thank the Center for Mining, Electro-Mechanical Research of Hanoi University of Mining and Geology (HUMG), Vietnam; the engineers and leaders of the Coc Sau open-pit coal mine, Quang Ninh province, Vietnam for their help and cooperation.

**Conflicts of Interest:** The authors declare no conflict of interest.

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
