# Peer review of "Estimating PM10 Concentration from Drilling Operations in Open-Pit Mines Using an Assembly of SVR and PSO"

_applsci, doi:10.3390/app9142806_

Round 1
Reviewer 1 Report
The authors have documented their AI-based machine leaning methods for estimating PM10 concentration from drilling operations in open-pit mines.The paper is detailed, well written, and enjoyable to read. The following are some suggestions for the authors to consider.
Suggestion 1:
Would the authors consider amending the word "advanced" in the title, "Estimating PM10 concentration from drilling operations in open-pit mines using advanced artificial intelligence techniques" to another choice of words, for example, "novel" or "assembly of SVR and PSO" or "hybrid SVR-PSO"?
The reason is, some readers out there who are very familiar with AI might point out that some of the latest AI techniques are Deep Learning (DL), or Reinforcement Learning (RL), or Recurrent Neural Networks (RNN), or Convolutional Neural Networks (CNN), or Generative Adversarial Networks (GAN). Including SVR-PSO in your title would also make it easier for readers to find and cite your work.
Suggestion 2:
Before line 115, can the authors please consider adding a section or paragraph to explain their rationale for creating their new hybrid model based on Support Vector Regression (SVR) and Particle Swarm Optimization (PSO)?
For example, out of so many other AI-related techniques, such as deep learning (DL), or reinforcement learning (RL), or recurrent neural networks (RNN), or convolutional neural networks (CNN), or generative adversarial networks (GAN), why were SVR and PSO chosen? For example, was it because of their unique suitability to analyzing this particular form of dust from open-pit mines, et cetera? This would inform readers who might not be familiar with the strengths of SVR and PSO to also consider using them to analyze PM10 dust in similar mining situations.
Suggestion 3:
In line 201, please change to font size 10 for the words "In this section, the RF was presented brief."
Thank you for considering these suggestions. I hope the authors may find them useful.
Reviewer 2 Report
Dear Authors,
Some comments hereafter:
In your paper you have considered and investigated several AI techniques. Why not ANN? They are one of the most representative techniques in ML domain and there are previous works using it (as you wrote in section 1). Is there a specific reason for that?
For table 2 (page 6) can you add more details, especially for the variables you indicated? In the text or directly in the table capture.
I think more details in figure 5 are necessary as well, especially for the different plots you presented (e.g. what each sub-figure means).
Can you indicate what is represented in Ox and Oy for figure 6 on page9?
In addition, be careful, because you indicated this figure as figure 5, which is not. The actual figure 5 is on page 8.
Consequently, you should re-number all the figures from here on (and, accordingly, modifying the corresponding labels in the text).
On page 10, lines 201-202, you should check the format.
Can you shortly described the figure on page 11 (e.g., what the numbers in the structure represent).
In the list of page 12, point 5), what r1 and r2 exactly mean?
Why have you decided to use PSO as optimization techniques for the hyper-parameters of SVR and not, for example, the Bayesian approach?
How did you select RMSE and R as metrics for the evaluation?
Idem for the first figure on page 15: can you add more details and info? I think more comments can help the reader; otherwise, it is not always easy to follow and understand the text.
The table on page 16 is not number 2 (it is on page 6). Maybe it is better to check all the numbers.
In this table, it would be better to make the hyper-parameters explicit.
In the figure on page 19, in the sub-plots (especially for the different PSO), there is one particular point, which is farer from the straight line (maybe providing the big error). Can you comment on that?
In the table on the same page (p. 19) more explanation of the parameters and symbols can help.
Finally, in your conclusions, there are no future activities or next steps. Anything is planned on that?
I hope these can be helpful for you
